**Comment**

# We urgently need a culture of multi-operationalization in psychological research

### Dino Carpentras

Analysis of different operationalizations shows that many scientific results may be an artifact of the operationalization process. A culture of multi-operationalization may be needed for psychological research to develop valid knowledge.

Measurements of abstract constructs have been criticized for a very long time in the social sciences due to their theoretical limitations[1]. However, their extensive use in research and practical applications suggests a general consensus that they can still yield valuable knowledge. Recent work emphasized the possibility that results obtained from such measurements of abstract constructs may be an artifact of the process of operationalization[2].

The present piece follows the definition in the American Psychological Association Dictionary of Psychology of "operational definition" as "a description of something in terms of the operations (procedures, actions, or processes) by which it could be observed and measured" and of "operationalization" as the process of creating such definition[3]. Thus, the operationalization of an abstract construct encompasses all decisions related to data collection, such as formulating items and response options, as well as to the data processing phase, such as the choice of analysis methods and how to handle outliers. Research questions involving constructs can be operationalized as well, by providing an operational definition of the involved constructs and by selecting appropriate tests.

One of the best examples of the impact of the operationalization phase has been provided by Breznau et al.[4] who asked multiple scientists to pursue the same research question on the same dataset. Results varied with the operationalizations employed: roughly 58% of the analyses produced null results, 25% significantly negative, and 17% significantly positive results. Only a tiny fraction of this variance was explained by aspects such as researcher's expertise.

## How we forgot about operationalization

Over the years, we have developed multiple tools for improving psychological measurements and the results derived from them. The entire branch of robust statistics is dedicated to dealing with problems like small samples and the presence of outliers. Theories like classical test theory and item response theory also provide foundations on how to obtain a latent variable from raw data and the current focus on replication helps to identify spurious results in the literature[5].

While these processes explore many fundamental aspects of measurements and statistical tests, they mostly neglect the role of operationalization. Indeed, even by fulfilling all the requirements for producing a valid measurement, the process of operationalization can be carried out in many different ways[4]. In the case of ordinal data, theory also allows the application of arbitrary non-linear transformations as long as they do not affect the ordering[6], and indeed, such transformations are often used for purposes such as correcting skewness. This increases even

more the possible choices that can be performed during the operationalization stage, and it raises the question if these choices can affect scientific results.

Ignoring this aspect is equivalent to implicitly assuming that different operationalizations, if done correctly, would lead to equivalent results. Many of us still consider this to be a reasonable assumption, but the research on operationalization seems to suggest otherwise.

## The problem is not in the latent nature

Results of these studies are often confusing to most researchers as it is not clear how operationalizations of the same construct can lead to contrasting results. This confusion is worsened by the fact that abstract constructs are not directly observable and do not allow us to explore what is happening behind the numbers. However, it is important to understand that surprising results are not due to the latent nature of the constructs, but naturally appear when we try to operationalize weakly defined concepts. To clarify the nature of the problem let's consider an example which does not relate to any latent construct: counting the number of potato chips in a bag. Indeed, similar to many abstract constructs, there is no operational definition of chips, leaving room for multiple operationalizations.

Initially, this task seems straightforward, as we possess an intuitive understanding of what constitutes a chip. However, complications arise when faced with broken chips. How should they be counted?

For the sake of the example, consider two scientists named Alice and Bob employing two different operationalizations. Alice may decide to count every individual piece in the bag, while Bob may choose to count only pieces which are above a threshold size. Clearly this produces different results while measuring the same construct.

Some readers may be tempted to think of this as a "measurement error." However, we could have an error only when there is a correct and unique way to count a broken chip, while this is not the case. Furthermore, this has nothing to do with statics and sampling as the scientists are assessing the same object.

We can also think of what would happen if they would run an experiment to explore the relationship between the number of chips and another variable, such as the pleasure of eating from that bag. Let's also suppose that people prefer eating full chips and that some bags have many broken chips while others have mostly big chips. In a similar situation, Alice and Bob are likely to observe opposite correlations, as illustrated in Fig. 1.

Finally, we can see how operationalization can affect the results even dynamically. If we start eating chips from a bag, both scientists will observe a decrease in the number of chips. Instead, if someone starts crushing the chips in the bag, Alice will measure an increasing number of chips and Bob the exact opposite.

In an example more relevant to the literature, Schweinsberg et al. recruited many analysts to test the hypothesis that "higher status participants are more verbose than lower status participants"[7]. Also in this case, all analysts had access to the same dataset while having complete freedom on the operationalization process. Indeed, some operationalized verbosity as

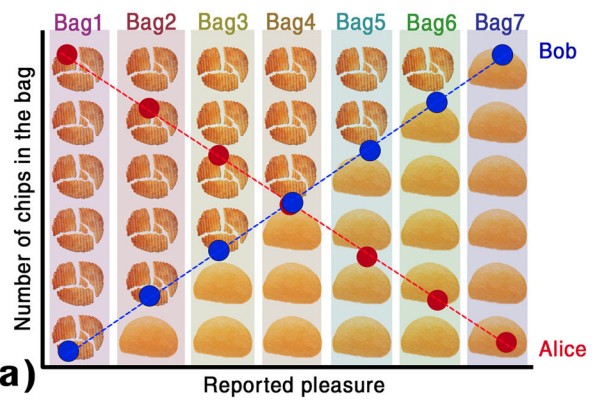
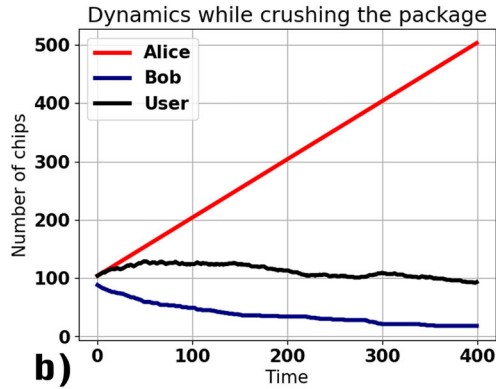

**Fig. 1 | Chips measurements under different operationalizations. a** Illustration of the regression analysis conducted by Alice and Bob on the same bags of chips. Each colored column represents a different bag. **b** Simulations of crushing the bag of chips.

The reader may repeat the dynamic experiment of Alice and Bob with varying operationalizations at the following link: https://www.dinocarp.com/chips-simulation/.

---

## Box 1 | The don'ts and dos of multi-operationalizaton

**Do Not:**

1. Do not confuse a construct with its operationalization. Even if the operationalization may show results which are statistically significant, it does not mean that you can generalize such result to the construct.

2. Do not assume equivalence of operationalizations, even if it has been shown in some cases. Different operationalizations may appear to be equivalent under certain conditions, as in the example of eating chips, while they may produce very different results in other situations, as when breaking the chips.

3. Do not assume that pre-registration and replication would automatically solve this problem. Indeed, both can be done by using only a single operationalization, therefore offering no insights on the result of other measurements. However, the two procedures can be easily coupled with the purpose of exploring different operationalizations.

**Do**

1. Explore the impact of different measurement instruments for the same construct. This can be achieved by asking multiple scientists to design them independently, showing different "practical" interpretations of the same construct.

2. Explore the consequences of different data processing methods. One way to perform this action consists in asking multiple scientists to work independently on the same dataset, as done in Breznau et al.[4]. Faster methods are possible as well, such as using code to automatically implement multiple models[10] or by testing the effect of non-linear transformations when using ordinal data.

3. Generalize results to the construct only if they are consistent across multiple operationalizations. If mixed results are obtained, this may be evidence that the explored constructs are just too complex or that our research question was too vague (Fig. 2).

---

the number of words in a comment, others as the number of characters, and some as the number of conversations one has participated in. Similar choices were also made for operationalizing "status" as well as for choosing the statistical model and possible covariates. These decisions may seem minor technical choices that should have only a marginal effect on the final outcome. However, as in the chips example, different operationalizations resulted in completely opposite results.

### Operationalizations are not dimensions

In the social sciences, a broad construct such as polarization is often divided into multiple dimensions, such as ideological consistency and affective polarization[8]. Furthermore, such dimensions are not supposed to be consistent with each other, thus potentially producing different results.

This situation may look very similar to what discussed above and may push some to confuse operationalizations with dimensions. However, it is important to notice that every dimension can still be operationalized in many ways. Thus, breaking a big concept into multiple dimensions can clearly improve its understanding and remove some conflicting results, but it is still insufficient to solve the problems of the operationalization process.

These considerations may feel disheartening to some, suggesting that every claim involving abstract constructs can be proven both true and false depending on the chosen operationalization. However, we need to stress that the literature on the topic has only focused on proving that there is a problem, but not on exploring its impact on most published research. These aspects will need to be explored in the future stream of research.

### Towards a culture of multi-operationalization

Many have criticized the problems of psychological measurements, especially in relationship to the vagueness of constructs, and have proposed solutions such as substantive theories or formalisms based on physical measurements of psychological phenomena[9]. While similar approaches may become the standard in the future, I believe that, for now, we should consider that reducing a complex construct to a single measurement might be just too simplistic. Indeed, the variety of results we obtain from different operationalizations may not be an error, but maybe just a feature of the complex nature of what we are studying.

An intuitive way to understand this issue through the parable of the blind people appraising an elephant. In this tale, every person touches a

**Fig. 2 | Relationships between construct and its operationalizations. a** Classic identification between construct and its operationalization. **b** How results can be generalized to the construct through multi-operationalization.

different area of the animal to understand what an elephant is. The one touching the tusks thinks the elephant is spear-shaped, while the one touching the leg thinks of it in the shape of a tree. The conflicting results in this case are due to the fact that the analysis tool cannot analyze the entire object, but only a fraction of it.

In a similar way, some constructs may just be too complex to be reduced to a single measurement. However, instead of trying to simplify abstract constructs to make them measurable, I urge exploring the consequences of their intricacy. Instead of fighting against complexity, I suggest embracing it. Overall, I propose developing a culture of multi-operationalization. By this I mean that we should not limit our analysis to one operationalization, but explore many of them to better understand the properties of constructs. In Box 1 and Fig. 2, I outline some actions we may take for starting to develop such a culture.

These guidelines should not be considered in a dogmatic way, I simply consider them a starting point for future studies. Indeed, more research will need to clarify the intricacies of the operationalization process. Future studies will need to explore how much of the published research can be reversed by equally valid measurements. Some may even develop new theories and combine them with statistics to standardize the process of multi-operationalization.

Overall, I do not think that the impact of operationalization should be seen in a negative way, but as a possibility to deepen our understanding of constructs and strengthen the quality of our research. Indeed, we have already developed a solid understanding of statistics and integrated the practice of pre-registration, so what is stopping us from developing a culture of multi-operationalization?

**Dino Carpentras** [ID] [✉]
ETH Zürich, Computational Social Science, Stampfenbachstrasse 48, 8092 Zürich, Switzerland. [✉]e-mail: dino.carpentras@gess.ethz.ch

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

### Acknowledgements
The author is grateful for support by the project "CoCi: Co-Evolving City Life", which received funding from the European Research Council (ERC) under the European Union's Horizon 2020 research and innovation programme under grant agreement No. 833168. The author would also like to thank prof. Dirk Helbing and prof. Mike Quayle for the inspiring discussions on this topic.

### Author contributions
D.C. was responsible for all aspects of this manuscript.

### Funding

### Competing interests
The authors declare no competing interests.
