## [Peer Review File · Communications Psychology]

29th Nov 23

Dear Dr Carpentras,

Thank you for your patience during the editorial evaluation and peer-review process.

Your manuscript titled "We urgently need a culture of re-operationalization of psychological measurements" has now been seen by 2 reviewers, and I have included their comments at the end of this message.

The reviewers are in principle enthusiastic about your work. However, they also mention a number of concerns. Editorially, we likewise feel that the piece needs to be improved substantially, in particular with regard to the stringency of the argument, the persuasiveness of the metaphors (which need to be more tightly linked to examples from the field), and general presentational issues.

We are very interested in the possibility of publishing your Comment in Communications Psychology but would like to consider your response to a list of concerns in the form of a revised manuscript before we make a decision on publication.

To aid you with that task, I have included a marked-up version of your manuscript.

In sum, we invite you to revise your manuscript taking into account all reviewer and editor comments.

EDITORIAL POLICIES AND FORMATTING

You will find a complete list of formatting requirements following this link:
<https://www.nature.com/documents/commsj-style-formatting-checklist-comment.pdf>
Please use the checklist to prepare your manuscript for resubmission.

If you have any questions about any of our policies or formatting, please don't hesitate to contact me.

Please use the following link to submit your revised manuscript and a point-by-point response to the referees' comments (which should be in a separate document to any cover letter):
[link redacted]

** This url links to your confidential home page and associated information about manuscripts you

may have submitted or be reviewing for us. If you wish to forward this email to co-authors, please delete the link to your homepage first **

We hope to receive your revised paper within 4 weeks; please let us know if you aren't able to submit it within this time so that we can discuss how best to proceed. If we don't hear from you, and the revision process takes significantly longer, we may close your file.

We understand that due to the current global situation, the time required for revision may be longer than usual. We would appreciate it if you could keep us informed about an estimated timescale for resubmission, to facilitate our planning. Of course, if you are unable to estimate, we are happy to accommodate necessary extensions nevertheless.

Please do not hesitate to contact me if you have any questions or would like to discuss these revisions further. We look forward to seeing the revised manuscript and thank you for the opportunity to review your work.

Best regards,

Marike Schiffer

Marike Schiffer, PhD
Chief Editor
Communications Psychology

REVIEWERS' COMMENTS:

Reviewer #1 (Remarks to the Author):

Thank you for the opportunity to review the manuscript entitled, "We urgently need a culture of re-operationalization of psychological measurements" for potential publication in Communications Psychology.

* What are the major claims of the paper?

In the manuscript, the author claims that results from studies are impacted by the operationalization process, that this process is not clearly defined, and that "re-operationalization" needs to occur. The paper starts by citing >>> and the study where researchers were asked to analyze the same data set and different results were found. The paper then goes on to explore how using different operational definitions for a potato chip leads to different results. The potato chip example is clear, as the operationalization is based on the definition of a potato chip. I am not clear on how the study with the same dataset could lead to different results based on different operationalization. This leads me to wonder if the term "operationalization" is clearly defined. Does it mean only operationalization of terms or is broader?

* Are these claims novel, and will they be of interest to others in the community? If the conclusions are not original, it would be helpful if you could provide relevant references.

As far as I know, these claims are novel as the importance of operationalization in research studies is not clearly addressed or discussed in many courses, books, or articles. Thus, this paper could add to

the literature.

* Is the work convincing, and if not, what further evidence would be required to strengthen the conclusions?

It would be helpful if the author defined operationalization and re-operationalization. I tried to utilize the webpage that is included in the paper and struggled to understand all the components. More explanation of the webpage would definitely be beneficial.

* On a more subjective note, do you feel that the paper will influence thinking in the field in terms of either conceptual understanding or technological capability? Please feel free to raise any further questions and concerns about the paper.

The paper would influence thinking in the field. The foundation of the paper, that researchers need to operationalize their terms, methods, etc. is an important concept and should be addressed as it would lead to more rigorous and replicable results.

Reviewer #2 (Remarks to the Author):

In "We urgently need a culture of re-operationalization of psychological measurements" the author argues for re-operationalization of psychological constructs.

I am of the opinion that all urgent calls to re-think measurement in psychology warrant attention, in that sense the commentary would be very welcome. I do have some remarks about the content, perhaps somewhat ironically about the liberal use of the term operationalization:

- The title mentions re-operationalization of psychological *measurements*... I think this is incorrect, because "operationalization refers to the process of turning abstract concepts or ideas into observable and measurable phenomena".

- Ideally, the term to use would be the *operational definition* of a concept, which contrasts the *theoretical definition* of a concept. One theoretically defined concept may have different operationalizations (e.g. stress measured as a self-report or physiological variable). The problem for psychology is that the theoretical definition is often missing (or is purely verbal) and we have no way to understand why different operationalizations of the same theoretical concept may lead to different outcomes, or how to connect them.

A second point is about measurement itself; the author doesn't really state what it means to perform a measurement of a psychological variable:

- Recently Borgstede & Eggert (2023) argued psychological measurement only makes sense in the context of substantive theories that can predict measurement outcomes in different contexts (just like in physics). In a commentary on this paper, it is pointed out that there isn't really a theory of the process of physical measurement of psychological variables (Hasselmann, 2023) and some suggestions are made on how to formally regard psychological measurement. I believe the author should add some text to explain their position in this debate, do they consider measurement to resemble classical physical measurement, or something else?

Borgstede M., Eggert F. (2023). Squaring the circle: From latent variables to theory-based measurement. *Theory & Psychology*, 33(1), 118–137. <https://doi.org/10.1177/09593543221127985>

Hasselmann, F. (2023). Going round in squares: Theory-based measurement requires a theory of measurement. *Theory & Psychology*, 33(1), 145-152. <https://doi.org/10.1177/09593543221131511>

I would really like to thank the reviewers and editor who spent time and energies for providing very useful comments. I believe the article has strongly improved thanks to these new perspectives.

The major changes in the manuscript have been:

- The introduction of definitions of the terms used allowing for a clearer and more fluent manuscript (e.g. “operationalization”)
- Clearer use of how the examples relate to the relevant literature
- Substantial expansion of the final section to be more “forward looking,” including some recommendations.

In the following I present the reviewers comments, highlighted in grey, and my responses.

REVIEWERS' COMMENTS:

Reviewer #1 (Remarks to the Author):

Thank you for the opportunity to review the manuscript entitled, “We urgently need a culture of re-operationalization of psychological measurements” for potential publication in *Communications Psychology*.

* What are the major claims of the paper?

In the manuscript, the author claims that results from studies are impacted by the operationalization process, that this process is not clearly defined, and that “re-operationalization” needs to occur. The paper starts by citing >>> and the study where researchers were asked to analyze the same data set and different results were found. The paper then goes on to explore how using different operational definitions for a potato chip leads to different results. The potato chip example is clear, as the operationalization is based on the definition of a potato chip. I am not clear on how the study with the same dataset could lead to different results based on different operationalization. This leads me to wonder if the term “operationalization” is clearly defined. Does it mean only operationalization of terms or is broader?

Regarding the example the reviewer is mentioning (i.e. “how the study with the same dataset could lead to different results based on different operationalization”) here the different results are due to different ways to process the data. For instance, which items should be considered, which criteria to use for exclusion, etc.

I have now made this clear by providing a clear and more detailed definition of operationalization. I also made clear how this is related to research questions and especially, to the data processing phase.

Indeed, the second paragraph in the introduction now reads:

The APA Dictionary of Psychology explains “operational definition” as “a description of something in terms of the operations (procedures, actions, or processes) by which it could be observed and measured” and “operationalization” as the process of creating such definition⁸. Thus, the operationalization of an abstract construct encompasses all decisions related to the data collection, such as formulating items and response options, as well as to the data processing phase, such as the choice of analysis methods and how to handle outliers. Research questions involving constructs can be operationalized as well, by providing an operational definition of the involved constructs and by selecting appropriate tests.

* Are these claims novel, and will they be of interest to others in the community? If the conclusions are not original, it would be helpful if you could provide relevant references. As far as I know, these claims are novel as the importance of operationalization in research studies is not clearly addressed or discussed in many courses, books, or articles. Thus, this paper could add to the literature.

* Is the work convincing, and if not, what further evidence would be required to strengthen the conclusions?

It would be helpful if the author defined operationalization and re-operationalization. I tried to utilize the webpage that is included in the paper and struggled to understand all the components. More explanation of the webpage would definitely be beneficial.

Besides adding a definition of “operationalization” I also added an explanation of “multi-operationalization”. Indeed, I opted for this term (instead of re-operationalization), as I believe it better expresses the idea I would like to convey. The second paragraph of the last section now reads:

[...] Overall, I propose developing a culture of multi-operationalization. With this I mean that we should not limit our analysis to one operationalization, but explore many of them to better understand the properties of constructs.

Regarding the website, I have now added an entire guide detailing what the simulation is doing, what are the different sliders and how to use everything. It can be found at:

<https://www.dinocarp.com/chips-simulation/>

The same link has also been updated in the main text.

* On a more subjective note, do you feel that the paper will influence thinking in the field in terms of either conceptual understanding or technological capability? Please feel free to raise any further questions and concerns about the paper.

The paper would influence thinking in the field. The foundation of the paper, that researchers need to operationalize their terms, methods, etc. is an important concept and should be addressed as it would lead to more rigorous and replicable results.

Thank you very much again for your inspiring words and for your insightful feedback.

Reviewer #2 (Remarks to the Author):

In "We urgently need a culture of re-operationalization of psychological measurements" the author argues for re-operationalization of psychological constructs.

I am of the opinion that all urgent calls to re-think measurement in psychology warrant attention, in that sense the commentary would be very welcome. I do have some remarks about the content, perhaps somewhat ironically about the liberal use of the term operationalization:

- The title mentions re-operationalization of psychological *measurements*... I think this is incorrect, because "operationalization refers to the process of turning abstract concepts or ideas into observable and measurable phenomena".

The reviewer is correct. I have now changed the title avoiding this misuse of the term. The title now reads:

We urgently need a culture of multi-operationalization in psychological research

- Ideally, the term to use would be the *operational definition* of a concept, which contrasts the *theoretical definition* of a concept. One theoretically defined concept may have different operationalizations (e.g. stress measured as a self-report or physiological variable). The problem for psychology is that the theoretical definition is often missing (or is purely verbal) and we have no way to understand why different operationalizations of the same theoretical concept may lead to different outcomes, or how to connect them.

I have also included the term of operational definition, both because of the good point raised by the reviewer, but also because the APA dictionary of psychology defines "operationalization" through "operational definition". I believe that by discussing both, it should be much clearer how these terms relate to each other, and how they are used during the article.

The second paragraph of the introduction now reads:

The APA Dictionary of Psychology explains "operational definition" as "a description of something in terms of the operations (procedures, actions, or processes) by which it could be observed and measured" and "operationalization" as the process of creating such definition⁸. Thus, the operationalization of an abstract construct encompasses all decisions related to the data collection, such as formulating items and response options, as well as to the data processing phase, such as the choice of analysis methods and how to handle outliers. Research questions involving constructs can be operationalized as well, by providing an operational definition of the involved constructs and by selecting appropriate tests.

A second point is about measurement itself; the author doesn't really state what it means to perform a measurement of a psychological variable: Recently Borgstede & Eggert (2023) argued psychological measurement only makes sense in the context of substantive theories that can predict measurement outcomes in different contexts (just like in physics). In a commentary on this paper, it is pointed out that there isn't really a theory of the process of physical measurement of psychological variables (Hasselmann, 2023) and some suggestions are made on how to formally regard psychological measurement. I believe the author should add some text to explain their position in this debate, do they consider measurement to resemble classical physical measurement, or something else?

Borgstede M., Eggert F. (2023). Squaring the circle: From latent variables to theory-based measurement. *Theory & Psychology*, 33(1), 118–137. <https://doi.org/10.1177/09593543221127985>

Hasselmann, F. (2023). Going round in squares: Theory-based measurement requires a theory of measurement. *Theory & Psychology*, 33(1), 145–152. <https://doi.org/10.1177/09593543221131511>

I really would like to thank the reviewer for such an interesting comment. It is important that a full explanation of what it means “to measure” would be rather long and complex. Within the scope of this article we initially focused on “measurements” as the results of the operationalization process (see the definition of operational definition above and its relationship to the concept of measurement). But most importantly, the articles and discussion suggested by the reviewer has now become the starting point for the conclusion section.

Indeed, the first paragraph of the conclusion now reads:

Many have criticized the problems of psychological measurements, especially in relationship to the vagueness of constructs¹⁵, and have proposed solutions such as substantive theories¹⁵ or formalisms based on physical measurements of psychological phenomena¹⁶. While similar approaches may become the standard in the future, I believe that, for now, we should consider that reducing a complex construct to a single measurement might be just too simplistic. Indeed, the variety of results we obtain from different operationalizations may not be an error, or even a problem to solve, but maybe just a feature of the multifaceted nature of what we are studying.

29th Jan 24

Dear Dr Carpentras,

Thank you for your patience during the editorial evaluation.

Your edits significantly improve the Comment and we remain very interested in the possibility of publishing your Comment in Communications Psychology. However, some additional revisions are strictly necessary and we would like to consider your response to a list of concerns in the form of a revised manuscript before we make a final decision on publication.

To aid you with that task, I have included a marked-up version of your manuscript. I attach both a clean copy and one that contains your previous edits and mine, so that you can trace what parts of the text needed to be revised.

EDITORIAL POLICIES AND FORMATTING

You will find a complete list of formatting requirements following this link:

<https://www.nature.com/documents/commsj-style-formatting-checklist-review-perspective.pdf>

Please use the checklist to prepare your manuscript for resubmission.

* TRANSPARENT PEER REVIEW: Communications Psychology uses a transparent peer review system. This means that we publish the editorial decision letters including Reviewers' comments to the authors and the author rebuttal letters online as a supplementary peer review file. However, on author request, confidential information and data can be removed from the published reviewer reports and rebuttal letters prior to publication. If your manuscript has been previously reviewed at another journal, those Reviewers' comments would not form part of the published peer review file.

If you have any questions about any of our policies or formatting, please don't hesitate to contact me.

Please use the following link to submit your revised manuscript

[link redacted]

We hope to receive your revised paper within 4 weeks; please let us know if you aren't able to submit it within this time so that we can discuss how best to proceed. If we don't hear from you, and the revision process takes significantly longer, we may close your file.

Please do not hesitate to contact me if you have any questions or would like to discuss these revisions further. We look forward to seeing the revised manuscript and thank you for the

opportunity to review your work.

Best wishes,
Marika

Marika Schiffer, PhD
Chief Editor
Communications Psychology

12th Mar 24

Dear Dino,

Your Comment titled "We urgently need a culture of multi-operationalization in psychological research". I am delighted to say that we are happy, in principle, to publish it in Communications Psychology under a Creative Commons 'CC BY' open access license.

We will not send your revised paper for further review if, in the editors' If the revised paper is in Communications Psychology format, in an accessible style, and of appropriate length, we shall accept it for publication immediately.

EDITORIAL REQUESTS:

* Please move the sentence, "The reader may repeat the dynamic experiment of Alice and Bob with varying operationalizations at the following link: <https://www.dinocarp.com/chips-simulation/> " to the Figure legend of Figure 1 and remove it from the main text.

* Please check whether your manuscript contains third-party images, such as figures from the literature, stock photos, clip art or commercial satellite and map data. If any of the display items in your manuscript (figures, tables, boxes or movies) include images that are the same as, or are adaptations of, previously published images, please fill in the Third Party Rights Table, and return to us when you submit your revised manuscript. This information will enable us to obtain the necessary rights to re-use such material. If we are unable to obtain the necessary rights to use or adapt any of the material that you wish to use, we will contact you to discuss alternative options.

* Communications Psychology uses a transparent peer review system. On author request, confidential information and data can be removed from the published reviewer reports and rebuttal letters prior to publication. If you are concerned about the release of confidential data, please let us know specifically what information you would like to have removed. Please note that we cannot incorporate redactions for any other reasons.

*If you have not done so already, please alert me to any related manuscripts from your group that are under consideration or in press at other journals, or are being written up for submission to other journals (see www.nature.com/authors/editorial_policies/duplicate.html for details).

FORMATTING GUIDELINES:

You will find a complete list of formatting requirements following this link:

<https://www.nature.com/documents/commsj-style-formatting-checklist-comment.pdf>

Please use the checklist to prepare your manuscript for final submission. In the following, I also highlight some issues of particular importance.

** Figures

Please remove all figures from the main text and upload them individually, one figure per file. To ensure the swift processing of your paper please provide the highest quality, vector format, versions of your images (.ai, .eps, .psd) where available. Text and labelling should be in a separate layer to

enable editing during the production process. If vector files are not available then please supply the figures in whichever format they were compiled in and not saved as flat .jpeg or .TIFF files. If your artwork contains any photographic images, please ensure these are at least 300 dpi.

* Figures should be simple and informative — multi-part figures are best avoided.

* References

References appear as superscript Arabic numerals, in order of mention. The reference list mentions references in the numerical order in which they are mentioned in the main text. If a reference is cited more than once, the same number is used throughout the text and the reference receives a single entry in the reference list.

Only papers that have been published or accepted by a named publication should be in the reference list (preprints and citations of datasets are also permitted). Unpublished/Submitted research should not be included in the reference list; it should only be mentioned briefly and parenthetically in the main text. Note that no major arguments should rely on unpublished research.

Published conference abstracts and URLs for websites should be cited parenthetically in the text, not in the reference list.

Footnotes are not used.

* Competing interests

Please include a "Competing interests" statement after the References. Note that we ask authors to declare both financial and non-financial competing interests. For more details, see <https://www.nature.com/authors/policies/competing.html>. If you have no financial or non-financial competing interests, please state so: "The authors declare no competing interests."

SUBMISSION INFORMATION:

* Your paper will be accompanied by a two-sentence editor's summary, of between 250-300 characters, when it is published on our homepage. We will use the preface as the Summary.

In order to accept your paper, we require the following:

* A cover letter describing your response to our editorial requests.

* The final version of your text as a Word or TeX/LaTeX file, with any tables prepared using the Table menu in Word or the table environment in TeX/LaTeX and using the 'track changes' feature in Word.

* Production-quality versions of all figures, supplied as separate files. Photographic images should be

300 dpi in RGB format (.jpg, TIFF or native Photoshop format) and any labels/scale bars included in a separate layer from the image. Line art, graphs and schemes should be vector format (.ai, .eps, .pdf); Adobe Illustrator files are preferred and will minimize production time. Any chemical structures or schemes contained within figures should additionally be supplied as separate Chemdraw (.cdx) files.

At acceptance, the corresponding author will be required to complete an Open Access Licence to Publish on behalf of all authors, declare that all required third-party permissions have been obtained.

Please note that your paper cannot be sent for typesetting to our production team until we have received this information; **therefore, please ensure that you have this ready when submitting the final version of your manuscript.**

ORCID

Communications Psychology is committed to improving transparency in authorship. As part of our efforts in this direction, we are now requesting that all authors identified as 'corresponding author' create and link their Open Researcher and Contributor Identifier (ORCID) with their account on the Manuscript Tracking System (MTS) prior to acceptance. ORCID helps the scientific community achieve unambiguous attribution of all scholarly contributions. For more information please visit <http://www.springernature.com/orcid>

For all corresponding authors listed on the manuscript, please follow the instructions in the link below to link your ORCID to your account on our MTS before submitting the final version of the manuscript. If you do not yet have an ORCID you will be able to create one in minutes.

IMPORTANT: All authors identified as 'corresponding author' on the manuscript must follow these instructions. Non-corresponding authors do not have to link their ORCIDs but are encouraged to do so. Please note that it will not be possible to add/modify ORCIDs at proof. Thus, if they wish to have their ORCID added to the paper they must also follow the above procedure prior to acceptance.

To support ORCID's aims, we only allow a single ORCID identifier to be attached to one account. If you have any issues attaching an ORCID identifier to your MTS account, please contact the Platform Support Helpdesk.

[link redacted]

We hope to hear from you within two weeks; please let us know if the process may take longer.

Best regards,

Marika

Marika Schiffer, PhD

Chief Editor

Communications Psychology